# Breastfeeding and Inborn Errors of Amino Acid and Protein Metabolism: A Spreadsheet to Calculate Optimal Intake of Human Milk and Disease-Specific Formulas

**DOI:** 10.3390/nu15163566

**Published:** 2023-08-13

**Authors:** Isidro Vitoria-Miñana, María-Luz Couce, Domingo González-Lamuño, Mónica García-Peris, Patricia Correcher-Medina

**Affiliations:** 1Metabolic and Nutrition Unit, Hospital Universitari i Politècnic la Fe, 46026 Valencia, Spain; garcia_monpera@gva.es (M.G.-P.); correcher_pat@gva.es (P.C.-M.); 2Department of Pediatrics, University Clinical Hospital of Santiago de Compostela, 15704 Santiago de Compostela, Spain; maria.luz.couce.pico@sergas.es; 3IDIS-Health Research Institute of Santiago de Compostela, 15704 Santiago de Compostela, Spain; 4Centro de Investigación Biomédica en Red de Enfermedades Raras (CIBERER), Instituto Salud Carlos III, 28029 Madrid, Spain; 5MetabERN, Via Pozzuolo, 330, 33100 Udine, Italy; 6Faculty of Medicine, Santiago de Compostela University, 15704 Santiago de Compostela, Spain; 7Pediatric Nephrology and Metabolism, Hospital Universitario Marqués de Valdecilla, 39008 Santander, Spain; domingo.gonzalez-lamuno@unican.es; 8Research Institute Valdecilla (IDIVAL), University of Cantabria, 39007 Santander, Spain

**Keywords:** breastfeeding, diet therapy, human milk, inborn error of metabolism, phenylketonuria, tyrosinemia 1, glutaric aciduria 1, homocystinuria, maple syrup urine disease, urea cycle disorder

## Abstract

Human milk (HM) offers important nutritional benefits. However, except for phenylketonuria (PKU), there are little data on optimal levels of consumption of HM and a special formula free of disease-related amino acids (SF-AA) in infants with inborn errors of metabolism of amino acids and proteins (IEM-AA-P). We designed a spreadsheet to calculate the amounts of SF-AA and HM required to cover amino acid, protein, and energy needs in patients with the nine main IEM-AA-P in infants aged under 6 months. Upon entering the infant’s weight and the essential amino acid or intact protein requirements for the specific IEM, the spreadsheet calculates the corresponding required volume of HM based on the amino acid concentration in HM. Next, the theoretical daily fluid intake (typical range, 120–200 mL/kg/day) is entered, and the estimated daily fluid intake is calculated. The required daily volume of SF-AA is calculated as the difference between the total fluid intake value and the calculated volume of HM. The spreadsheet allows for the introduction of a range of requirements based on the patient’s metabolic status, and includes the option to calculate the required volume of expressed HM, which may be necessary in certain conditions such as MMA/PA and UCD. In cases in which breastfeeding on demand is feasible, the spreadsheet determines the daily amount of SF-AA divided over 6–8 feeds, assuming that SF-AA is administered first, followed by HM as needed. Intake data calculated by the spreadsheet should be evaluated in conjunction with data from clinical and nutritional analyses, which provide a comprehensive understanding of the patient’s nutritional status and help guide individualized dietary management for the specific IEM.

## 1. Introduction

Human milk (HM) has multiple beneficial effects in infants, including providing polyunsaturated fatty acids, bioavailable iron, non-protein nitrogen (e.g., lactoferrin, nucleotides, polyamines), oligosaccharides, and immune factors (e.g., immunoglobulins, cytokines, and growth factors), as well as promoting mother-child attachment via breastfeeding [1,2]. HM has several additional benefits in infants with inborn errors of metabolism (IEM): it is associated with reduced incidence of diarrhea and respiratory infections [3], reduced propionate formation at the intestine [4], and, in the long term, better cognitive development and a lower risk of non-communicable chronic diseases such as high blood pressure and diabetes [5]. In the specific case of phenylketonuria (PKU), European guidelines recommend HM over standard infant formula (IF) as a source of phenylalanine (Phe) owing to its lower Phe content (46 mg/100 mL) [6]. In addition, breastfeeding on demand provides the mother with greater control over the feeding process.

HM is contraindicated in the following IEMs: galactosemia, severe forms of long-chain fatty acid β-oxidation disorders, and glucose-galactose malabsorption [7]. However, due to its lower protein and amino acid content compared with IF, its use has been proposed in combination with a special formula lacking the amino acid to be avoided (SF-AA), especially in PKU [8,9,10,11], probably because this was the first IEM included in newborn screening (NBS) programs in most countries. In addition to PKU, other infants with intermediary IEMs of amino acids and/or proteins (IEM-AA-P), including tyrosinemia I type 1 (TYR-1), classic homocystinuria (HCU), glutaric aciduria I (GA-I), maple syrup urine disease (MSUD), isovaleric aciduria (IVA), methylmalonic/propionic aciduria (MMA/PA), and urea cycle defects (UCD), could also benefit from HM owing to its lower protein content compared with IF. However, accumulated experience in these IEMs is limited, and available data often correspond to short durations of HM administration [12].

Adequate intake of protein and amino acids is necessary to achieve anabolism with normal growth. Excessive protein restriction can lead to protein-energy malnutrition with poor growth, while excessive protein intake can trigger metabolic decompensation. While total protein is a key consideration in IEM-AA-P, an inadequate balance of individual amino acid concentrations can also negatively affect absorption, protein synthesis, and brain concentrations of essential amino acids [13,14]. For patients with this type of IEM, it is essential to provide sufficient energy throughout the day to achieve an effective anabolic state. For these reasons, it is very important to correctly calculate the amounts of SF-AA and HM to be given to infants and to compare the total intake of amino acids, protein, and energy with those indicated in published recommendations.

As breast milk macronutrient and mineral content cannot be directly analyzed for individual mothers and infants, mean values are useful to individualize dietary requirements in different IEMs [15]. We have developed a spreadsheet to calculate the amounts of SF-AA and HM that infants with intermediary IEMs should take daily to cover amino acid, protein, and energy needs without reaching toxic levels in order to facilitate the management of mixed breastfeeding in infants with IEMs. The study is limited to infants under 6 months of age, as this is the period for which HM is recommended by the WHO; after 6 months, additional calculations will be required to determine intake of other dietary components (mainly fruits and vegetables).

## 2. Materials and Methods

### 2.1. Study Design and Population

This virtual study was carried out in 2023 with the aim of developing a calculation spreadsheet for infants aged 0–6 months with IEM-AA-P, with the participation of healthcare professionals from tertiary centers with experience in these IEMs (Appendix A).

### 2.2. Methodology

Specific spreadsheets were developed for each of the following IEM-AA-P: PKU, TYR-1, GA-I, HCU, MSUD, IVA, MMA/PA, and UCD.

Body weight (in kg) and the required intake of the essential amino acid involved in the pathology or “limiting amino acid” (Phe in PKU; tyrosine (Tyr) in TYR-1; lysine (Lys) in GA-1; methionine (Met) in HCU; leucine (Leu), valine (Val), isoleucine (Ile) in MSUD; and Leu in (IVA)) in mg/kg/d or in mg/d are entered into the spreadsheet. The recommended amount of the limiting amino acid lies in a range. Based on the concentration of amino acids in HM (Table 1) [16], the volume of HM needed to provide the recommended amount of the “limiting amino acid” is calculated. The theoretical daily intake (120–200 mL/kg/d) is then entered, and the estimated daily fluid intake is calculated. The daily volume of SF-AA is calculated as the difference between the total fluid intake value and the calculated HM volume. The spreadsheet output includes a summary indicating the daily intake of the essential limiting AA, total energy, and both intact protein (from HM) and synthetic protein (from SF-AA). These values can be compared with the recommendations in IEM guidelines [6,17,18,19,20,21,22,23,24,25,26] for infants aged 0–6 months (Table 2). Finally, based on the total amount of SF-AA, a practical recommendation is made regarding the volume of SF-AA that the infant should consume before 6 or 8 breastfeeding feeds. The discordant energy requirements for distinct IEMs are due to the fact that recommendations from the corresponding protocols or guidelines were respected.

In the case of MMA/PA and UCD, the daily amount of HM was calculated based on the whole protein requirements specific to each IEM group, following the recommendations in the literature. However, it is important to note that the recommended amount can vary over a range of values depending on the patient’s clinical situation or prior analytical data. In these IEM-AA-P, the mother often has to express and measure the quantity of HM, especially during the first weeks after birth. The total amount of HM required is indicated in the spreadsheet, which also enables the calculation of the required amount spread over 6 or 8 daily feeds. Next, the estimated total daily fluid intake is entered, and the amount of special formula without Met, Thr, Val, and Ile (SF-Met-Thr-Val-Ile) for MMA/PA or special formula with essential amino acids (SF-AAE) for UCD is calculated per day, divided over 6–8 feeds.

The flow chart in Figure 1 depicts the proposed sequence of calculations.

The formulas of the spreadsheet are not displayed in each IEM to avoid inadvertent errors and modifications. There is a sheet entitled “formulas”, in which the calculations used in each of the cells are indicated. The spreadsheet can be obtained in Appendix A.

## 3. Results

### 3.1. Scenario 1: Special Formula Lacking the Limiting Amino Acid Is Available

In infants with inborn errors of metabolism of amino acids (IEM-AAs) (PKU, TYR-1, AG-I, HCU, MSUD, and IVA), body weight and the recommended amount of the limiting amino acid are entered to calculate the corresponding amount of HM. The theoretical daily intake (120–200 mL/kg/d) is then entered to estimate the daily fluid intake. The daily volume of SF-AA is calculated as the difference between the total fluid intake value and the calculated HM volume. For infants weighing >7 kg or aged >4 months, the theoretical daily intake is likely to be 120–150 mL/kg/d. In all cases, the recommended amount of the limiting amino acid can be adjusted within the recommended range based on the patient’s metabolic status, particularly in cases of MSUD and severe IVA. In these situations, the amount of HM obtained can serve as a guide for the mother to express a corresponding amount of HM.

### 3.2. Scenario 2: Total Amount of Protein Intake Must Be Calculated

There are two groups of IEM-AA-P for which it is necessary to calculate the total amount of protein rather than amino acids: MMA/PA and UCD. In these cases, after entering the infant’s weight, we input the intact protein requirements (in MMA/PA, the value range is 0.91–1.52 g/kg/day; in UCD, the value range is 0.8–1.1 g/kg/day). The user decides which value to enter based on the patient’s degree of metabolic stability. Using the resulting value, we calculate the volume of human milk (HM) containing that amount of protein. In both MMA/PA and UCD, due to the risk of metabolic decompensation associated with excessive protein intake from ad libitum HM feeding, we opted to limit the amount of HM to the calculated value. To this end, the mother must express milk using a breast pump. Our recommendation is to offer the calculated amount of HM over six or eight bottle feeds. This is followed by the calculated amount of bottle-feeding with SF-Met-Thr-Val-Ile (in MMA/PA) or synthetic formula amino acid supplement (SF-AAE) (in UCD). The spreadsheet divides the daily amount of HM and SF-Met-Thr-Val-Ile or SF-AAE over six or eight feeds, ensuring appropriate distribution throughout the day.

SF-AA volumes vary widely, depending on the weight, estimated intake, and disease in question. Thus, no assumptions can be made as to SF-AA amounts, expressed as a percentage of total intake. The great advantage of this spreadsheet is that it enables immediate calculation of results and allows for small variations.

## 4. Discussion

Here, we describe a tool that enables the calculation of the amount of HM and special formula required for infants aged 0–6 months with intermediary IEMs. While HM composition varies between individuals, mean values can provide useful estimates of milk content. The tool described here allows for a rapid calculation of the recommended intake for these diseases.

The dietary management of IEM-AA-P can be divided into three stages: management of the acute episode, adaptation to breastfeeding, and long-term management [7]. In patients who are symptomatic at diagnosis, the primary objectives during the acute phase are to maintain biochemical balance through detoxification techniques. These include treating hyperammonaemia in UCD, MMA/PA, and IVA; treating ketoacidosis in MMA/PA, IVA; specific treatments such as nitisinone in TIR-1; carnitine in GA-I or pyridoxine in HCYS; and administration of SF-AA to decrease levels of toxic amino acids in most IEM-AAP, as well as promoting anabolism through the administration of carbohydrates and lipids.

After the acute phase, patients can begin receiving intact protein from a calculated quantity of HM. Once metabolic control is well established, the expressed and measured HM can be followed by breastfeeding on demand, at the discretion of the healthcare team. Modular ingredients, such as special essential amino acid (EAA)-based feeds developed for IEMs and energy supplements containing glucose polymers and fats, can be used in combination with expressed breast milk or breastfeeding on demand.

Regular monitoring of clinical parameters, including general well-being, feeding records, body weight and height, and biochemical parameters such as quantitative plasma amino acids, serum ammonia, urinary ketones, albumin, prealbumin, hemogram, and organic acid analysis, should be conducted on a weekly or biweekly basis after discharge from the hospital. In some cases and specific IEM-AAs such as PKU or HCU, it may be considered to offer ad libitum breastfeeding for 24 h initially, followed by adjustment of the amounts of HM based on the results of analyses. This approach allows for a period of unrestricted breastfeeding to assess the infant’s response and tolerance to the intact protein in HM. After this initial period, the analytical results can guide the adjustment of HM amounts, ensuring that the amino acid levels are within the desired target ranges for optimal metabolic control.

There are limited tools available specifically designed to calculate the amount of HM and SF-AA that can be safely consumed by IEM-AA patients. Existing recommendations for conditions such as PKU, GA-I, MSUD, IVA, and HCU provided by organizations such as the British Inherited Metabolic Diseases Group (BIMDG) serve as valuable references [28]. The proposed spreadsheet described here is among the first tools designed specifically for this purpose. It aims to provide a convenient and practical means of calculating the appropriate volumes of HM and SF-AA tailored to the individual needs of infants with IEM-AAs. By utilizing this spreadsheet, healthcare professionals can have access to a tool specifically designed for these calculations, which can facilitate more accurate and personalized dietary management for patients with IEM-AAs. As research and understanding of IEM-AAs continue to evolve, it is likely that additional tools and resources will be developed to further support the calculation and management of HM and SF-AA intake.

For infants with PKU aged less than 6 months, the recommended daily Phe intake is 45–55 mg/kg/day in order to achieve plasma concentrations of 120–360 µmol/L [6]. The recommended intake of a special formula lacking Phe (SF-Phe) corresponds to 80–100 Kcal/kg/d and 0.8–1.3 g/kg/day of protein.

Only after neonatal diagnosis and stabilization of plasma Phe concentrations within this range through administration of SF-Phe can HM be used as a source of intact protein, provided that it contains 46 mg/100 mL Phe [29]. Initial Phe tolerance is established in early infancy by titrating the amount of HM or standard IF with blood Phe levels [6].

The use of HM in the dietary management of infants with PKU has been an area of considerable interest for several decades. One of the earliest experiences with HM in PKU was reported by McCabe et al. [30] in a study that included 18 infants diagnosed by neonatal screening between 1977 and 1984. The study found that infants with PKU who were fed with HM had similar outcomes to those who were fed with formula. Since then, accumulated experience in the use of HM for infants with PKU has contributed to a growing body of evidence. As research and clinical knowledge have advanced, healthcare professionals have gained further insight into the benefits and challenges of using HM in PKU management. This in turn has led to the development of guidelines and recommendations to guide the use of HM in conjunction with Phe-restricted diets and special amino acid-based formulas.

The most widely used method for HM administration in PKU is based on providing a measured volume of SF-Phe before each feed. This reduces the infant’s appetite, thereby decreasing the amount of HM and, consequently, Phe ingested [30,31]. Infants continue to feed on demand, varying the number of feeds from one day to the next, but SF-Phe is limited. Most breastfed infants take 6–8 daily feeds of Phe-free formula before breastfeeding. Therefore, our spreadsheet also provides bottle volumes depending on whether the infant feeds six or eight times per day. It is good practice to encourage parents to keep a record of daily feeds [32,33]. The method of administering SF-Phe before breastfeeding has shown that infants with PKU continue to breastfeed for an older age of HM compared to the alternative methods of alternating HM and SF-Phe or expressing milk and measuring the amount administered [34,35].

The number of feeds per day administered to PKU infants is based on Phe blood levels. Blood Phe concentrations should be checked twice weekly until they stabilize. The infant should be weighed weekly for at least the first 6 weeks. Frequent weighing helps ensure that the infant is gaining weight and, by inference, ingesting enough HM. Ideally, HM should be provided for at least 6 months [36].

The goal of nutritional therapy in TYR-1 patients treated with a low-tyrosine diet and nitisinone (NTBC) is twofold: to restrict Phe and Tyr intake and achieve normal growth and development. Most dietary Phe is hydroxylated to produce Tyr, so Phe intake must be restricted in the diets of affected patients. To this end, a reduced amount of intact dietary protein should be prescribed [18,37]. To fulfil protein, energy, and nutrient needs, it is necessary to use a formula containing mixtures of amino acids lacking Phe and Tyr (SF-Phe-Tyr). To achieve the most rapid decrease in Tyr in infants after diagnosis, SF-Phe-Tyr can be administered for up to 48 h. After entering the weight in the spreadsheet, a Tyr intake value is inputted based on the patient’s clinical and analytical status. This allows for the calculation of the volume of HM containing the specified amount of Tyr. Additionally, it is recommended to adjust the total protein intake to 120 Kcal/kg/day and 2.5–3.5 g/kg/day. Therefore, following diagnosis and stabilization of the patient with NTBC and a low Tyr diet (SF-Phe-Tyr), the low Tyr-Phe diet should be continued indefinitely with careful monitoring. This includes regular checks of physical growth, nutritional analysis, liver and renal function tests, amino acid levels (target plasma concentrations of Tyr are 200–400 μmol/L while Phe levels should correspond to normal plasma values), as well as monitoring of succinylacetone levels in plasma or urine and NTBC levels [38]. Two reported cases of tyrosinemia I involved infants who were breastfed, preceded by infant protein substitutes, for 22 and 23 weeks, respectively. In both cases, Phe levels were low, necessitating Phe supplementation [39].

In the case of GA-I, the latest international recommendations encourage HM on demand after administration of a calculated amount of amino acid formula lacking Lys and with low levels of tryptophan (Trp) (SF-Lys-Trp), analogous to PKU recommendations [19]. This procedure has been implemented in various clinical trials, demonstrating favorable clinical outcomes [40]. Additionally, the progression of five patients with GA I who were breastfed for 2 to 11 months, with parallel use of SF-Lys-Trp, has been documented. These patients exhibited normal lysine levels, appropriate physical development, and typical psychomotor development considering the nature of the disease [12,41,42]. The British Inherited Metabolic Diseases Group (BIMDG) also utilizes this method for calculating amounts of SF-Lys-Trp and suggests a fluid intake of 150, 175, or 200 mL/kg/d [43]. Following these guidelines, our calculation spreadsheet incorporates the recommended intake of 100 mg Lys/kg/day, 80–100 Kcal/kg/day, and 0.8–1.3 g/kg/day protein from SF-Lys-Trp. The concentration of Lys in human milk is set at 90 mg/100 mL. In the follow-up of these patients, it is important to assess their physical and psychomotor development and to include nutritional analysis and measurements of lysine, free carnitine, and glutaryl-carnitine levels [19].

In cases of HCU detected by NBS, it must first be established whether the HCU is pyridoxine-sensitive. This is achieved by administering pyridoxine and folic acid for 14 days, after which total homocysteine (Hcy) is measured and compared with a threshold of 50 μmol/L. Dietary management of HCU can be highly successful. It should be considered for all pyridoxine-unresponsive patients and as an additional treatment for individuals who are partially pyridoxine-responsive [21]. UK guidelines for infants diagnosed via NBS recommend discontinuing the supply of Met/ intact protein (from HM or IF) after performing the pyridoxine test and providing a Met-free formula (SF-Met) for 2–4 days to reduce Hcy levels [44]. Met in the form of HM or IF is then introduced, divided over several feeds, in conjunction with SF-Met [45]. The recommended starting concentration is 90–120 mg Met/day (30 mg/kg/d in infants weighing <3 kg). This recommendation is similar to that of the Spanish Association for the Study of Inborn Errors of Metabolism (AECOM) [20]. After entering the weight, we input the recommended daily intake of Met (between 90 and 120 mg), which is used to calculate the volume of breastfeeding that contains the specified amount. BIMDG advises restricting HM intake to approximately 50% by providing SF-Met (approximately 75 mL/kg/day) before HM and waiting for plasma Hcy and Met to decline within 7–12 days to the target range of 60–100 μmol/L for Hcy and to the normal physiological range for Met [21]. From that moment on, the spreadsheet would have greater value.

An international study documented the use of measured volumes of synthetic formula with SF-Met prior to HM intake in five infants with HCU. This approach aimed to limit the amount of HM and Met ingested by the infants [46]. Additionally, a case report described a patient with HCU who was breastfed until 18 months of age while maintaining normal homocysteine (Hcys) and methionine controls [47]. Another report described three HCU patients who received both HM and SF-Met for a period of 6 months, with an estimated daily fluid intake of 160 mL/kg/day. In one of these cases, breastfeeding had to be interrupted for 12 weeks due to insufficient SF-Met intake [48]. These studies highlight the importance of early diagnosis and appropriate management in HCU. They demonstrate different approaches to breastfeeding and the use of SF-Met in order to control methionine intake and maintain metabolic control. Monitoring of plasma Hcy and Met levels and growth is necessary in HCU patients, who require low Met intake to achieve adequate biochemical control.

In mild forms of MSUD, where tolerance to branched-chain amino acids is higher, breastfeeding can be possible. However, for patients with severe forms of the disease, breastfeeding becomes more challenging and requires careful monitoring. Guidelines for infants <6 months with MSUD recommend a Leu intake of 40–100 mg/kg/day, 95–145 Kcal/kg/day, and 2.5–3.5 g/kg/day of total protein [22]. In the spreadsheet, after entering the weight, an estimated value between 40 and 100 mg/kg/day should be inputted, depending on the patient’s clinical and analytical situation.

There have been reported cases of MSUD in which breastfeeding with expressed HM and supplementation were successfully implemented. For example, one study described a patient who consumed expressed HM, corresponding to 1.0 g/kg/day of intact protein, supplemented with 1.5 g/kg of synthetic formula amino acid (SF-AA) protein for 55 days. Subsequently, HM was offered on demand [12]. However, breastfeeding was discontinued at 4 months of age due to a twofold increase in plasma concentrations of branched-chain amino acids. Another study described a MSUD patient who was fed expressed breastmilk and an additional formula lacking branched-chain amino acids for 1 month. The patient showed adequate neurological and weight development, but breastfeeding was stopped due to insufficient milk production [41]. A third study described three cases in which infants alternated between HM for 3–11 months and bottles containing SF-AA and showed normal growth and branched-chain amino acid values [49]. These studies illustrate the varied experiences and challenges associated with breastfeeding for infants with MSUD. The decision to breastfeed should be made in consultation with healthcare professionals, taking into account the specific needs and responses of the individual infant, and including regular monitoring of health and metabolic status.

For IVA patients, the goal of treatment is to minimize isovaleric acid levels. Treatment consists of a low-protein diet with glycine and carnitine supplementation and should be started as soon as possible after birth. In many patients, normal amounts of protein are tolerated; in some cases, a special Leu-free amino acid (SF-Leu) formula is necessary, in which case excessive protein intake should be avoided. The recommended leucine intake, according to Knerr et al., is 50–140 mg/day [23]. The specific value for leucine intake can be determined based on the patient’s clinical and analytical situation. Gockay et al. described two cases of IVA. One patient was fed with expressed breast milk for 13 days and breastfeeding on demand for 1 month (due to hypogalactia), while the other was provided with breastfeeding on demand for 10 months. Both patients showed positive outcomes [50].

In MMA/PA, after initial stabilization in cases of decompensation with metabolic acidosis and/or hyperammonaemia, intake of intact protein in the first 6 months of life is recommended, which is equivalent to 60–100% of RDI requirements [50]. This translates into values of 0.90–1.52 g/kg/day [24], which is complemented with the SF-Met-Thr-Val-Ile formula to reach 1.52–1.82 g/kg/day total protein. In their review, Pichler et al. [41] include information on three patients with PA and two patients with MMA who received HM. Two of the PA patients were fed with expressed breast milk, while the remaining patients were breastfed on demand. In all cases, an amino acid mixture free from precursor amino acids was administered with a bottle immediately before on-demand breastfeeding. One issue observed during the period of breast milk feeding in this subgroup was transient poor weight gain in one MMA patient. In their survey, McDonald et al. [46] documented seven infants with PA and six with MMA who were breastfed for up to 16 weeks. Gockay et al. [42] included information on four patients with MMA, in which expressed HM was provided in two cases, followed by a switch to breastfeeding on demand after 1 week. These patients maintained HM feeding with good metabolic control for 8 and 24 months, respectively. The other two MMA patients exclusively received HM on demand and continued HM feeding for 12 and 21 months The study also describes a patient with PA who could only maintain HM for 3 months due to experiencing two metabolic decompensations, which led to the discontinuation of HM as an intact protein source. Using the spreadsheet provided here, it is possible to calculate the amount of HM that the mother should express, particularly in the days after initial metabolic stabilization. Additionally, the spreadsheet can determine the amount of SF-Met-Thr-Val-Ile that the infant should consume before transitioning to on-demand breastfeeding with HM.

The treatment of UCD involves restricting dietary protein, providing sufficient energy to prevent catabolism, and using nitrogen-scavenging drugs. According to McLeod [26], a recommended intake of 0.4–1.1 g/kg/day of SF-AAE and 0.8–1.1 g/kg/day of intact protein is suggested, resulting in a total protein intake range of 1.2–2.2 g/kg/day. In our spreadsheet, considering the high clinical variability of UCD, we can input a value in the range of 0.4–1.1 g/kg of intact protein, and the corresponding amount of HM will be calculated. In one published case of severe ornithine transcarbamylase (OTC) deficiency, HM was discontinued at 1.5 months due to hyperammonaemia associated with a respiratory infection. In that case, the newborn received expressed HM for 1 month [12]. Pichler et al. [41] described the response of three UCD patients (one case of OTC deficiency and two cases of arginase deficiency (AD)) who received HM for 2–9 months. In all cases, neurodevelopment, weight, and height were within appropriate limits for age, with no metabolic decompensations. Two patients were breastfed on demand (AD *n* = 1, OTC *n* = 1), while one patient received expressed breast milk (AD *n* = 1) [41]. The literature also includes a documented case of neonatal citrullinemia in which a controlled amount of HM (0.6 g/kg/day) was administered, with favorable outcomes [51]. These results emphasize the importance of determining the severity of the defect in each UCD patient and providing individualized treatment. In our opinion, in cases of UCD, such as MMA/PA and some severe cases of IVA, it is advisable to express HM and provide the calculated amount in a bottle. This can be supplemented with SF-AAE to make up the estimated fluid intake. We suggest calculating the total protein with the spreadsheet and, if it is excessive, supplementing the energy needs with lipid and/or carbohydrate modules, thus avoiding excessive SF-AAE intake. As the infant grows older, breastfeeding on demand can be considered in selected cases. Regular monitoring of ammonia and amino acid levels, along with nutritional assessments, should be carried out.

One of the great advantages of the spreadsheet described here is its versatility and the speed with which the proposed HM and SF-AA intake can be determined. For instance, in the case of a newborn with severe MSUD and a bodyweight of 3.5 kg, after the management phase of the acute episode, we could opt to introduce the minimum requirement of Leu (40 mg/kg/d). Using the spreadsheet, we can immediately see that this amount of 140 mg of Leu could be achieved with an intake of 121 mL/d HM, which also contains 1.3 g of protein, equivalent to 0.4 g of protein/kg. This quantity needs to be expressed and supplemented with SF-AA. Assuming an intake of 150 mL/kg, we could supplement with 404 mL of SF-AA, resulting in eight bottles containing 15 mL of HM and 51 mL of SF-AA that can be offered to the infant. Upon reviewing the spreadsheet, we can ensure that the intakes of Leu, Val, and Ile are within the lower limit. As the infant grows, with a weight of 5 kg and normal Val, Leu, and Ile values, we can calculate an intake of 80 mg Leu/kg, which can be achieved with 345 mL of HM. Considering a fluid intake of 150 mL/kg, this could be supplemented with 405 mL of SF-AA. By examining the bottom of the spreadsheet, we can confirm that the intake of protein, Val, Leu, Ile, and energy would fall within the recommended range. Moreover, we can observe that the amounts of HM and SF-AA in each feed are similar, suggesting that breastfeeding on demand after SF-AA feeding could be considered in this scenario.

The spreadsheet’s ability to rapidly provide personalized calculations and assess different scenarios makes it a valuable tool for well-informed decision-making in the dietary management of infants with IEM-AAs.

Some limitations of the present study should be noted. Patients with milder forms of the disorder will tolerate higher amino acid or protein intake and may only require a reduction in intact protein; therefore, they can take a higher volume of HM and a lower volume of special formula. Another limitation is that the spreadsheet calculations have been developed based on specific formulas available in Spain, although adaptation for use from other countries is also possible. This spreadsheet has been developed based on intake recommendations from relevant guidelines. However, it has not undergone extensive testing on a large population of patients. While it can serve as a helpful tool for calculating and estimating intake requirements, it is important to consider individual patient factors, clinical assessment results, and professional judgment when interpreting and implementing the output.

In summary, we have developed a tool that enables rapid calculation of HM requirements in infants with IEM-AAs. By entering the infant’s body weight and the requirements for the limiting amino acid or intact protein, the tool provides an estimate of the required HM volume. In certain conditions, such as MMA/PA, UCD, and severe cases of IVA, it may be more reliable to measure the volume of HM by expressing it. For other conditions, HM can be offered on demand after SF-AA feeding. The spreadsheet also allows for the calculation of the necessary volume of each of the six or eight daily SF-AA bottles based on the estimated total daily fluid intake. It is important to note that the values generated by the spreadsheet serve as general guidance only and must be individualized for each patient, taking into account the severity of their disorder. In cases of severe enzyme deficiency, caution should be exercised when interpreting the spreadsheet results, and adjustments should be made based on the specific biochemical parameters associated with each IEM-AA.

## Figures and Tables

**Figure 1 nutrients-15-03566-f001:**
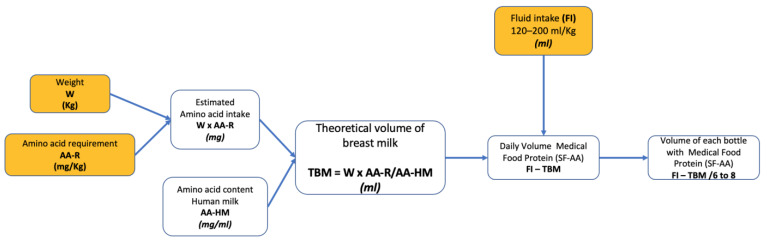
Flowchart for calculating the volume of each SF-AA. Data to be entered in the spreadsheet are shown in yellow. AA-R, amino acid requirement; AA-HM, amino acid content of human milk; FI, fluid intake; TBM, theoretical volume of breast milk; W, weight; FI, Fluid intake; SF-AA, special formula without one or more amino acids.

**Table 1 nutrients-15-03566-t001:** Estimated mean amino acid, protein, and energy content of human milk [16].

Nutritional Component	Per 100 mL
Kilocalories	70 Kcal
Protein	1.07 g
Phenylalanine	46 mg
Tyrosine	36 mg
Lysine	90 mg
Methionine	18 mg
Leucine	116 mg
Isoleucine	64 mg
Valine	83 mg

**Table 2 nutrients-15-03566-t002:** Recommended daily intake of limited essential amino acids or protein, and energy for each IEM [6,17,18,19,20,21,22,23,24,25,26].

IEM	Limited Essential AA/Protein	RDI (0–6 m)	Kcal/kg/d	Protein g/kg/d
PKU [6,17]	Phe	45–55 mg/kg/d	80–100	0.8–1.3 (SF-AA)
TYR-1 [18]	Tyr	95–275 mg/d (Tyr) 185–550 mg/d (Phe)	120	2.5–3.5 (TP)
GA- I [19]	Lys	100 mg/kg/d	80–100	0.8–1.3 (SF-AA)
HCU [20,21]	Met	90–120 mg/d	95–145	2.5–3.5 (TP)
MSUD [22]	Leu, Ile, Val	40–100 mg/d (Leu) 30–90 mg/d (Ile) 40–95 mg/d (Val)	95–145	2.5–3.5 (TP)
IVA [23]	Leu	50–140 mg/kg/d	90–120	2.5–3.0 (TP)
MMA/PA [24,25]	Protein	0.9–1.5 g/kg/d (IP)	72–109	1.5–1.8 g/kg/d (TP)
UCD [26,27]	Protein	0.8–1.1 g/kg/d (IP)	80–100	1.2–2.2 g/kg/d (TP)

SF-AA, special formula without one or more amino acids; GA-I, glutaric aciduria I; HCU, classic homocystinuria; IEM, inborn errors of metabolism; IVA, isovaleric aciduria; Leu, leucine; Lys, lysine; Met, methionine; MMA/PA, methylmalonic/propionic aciduria, MSUD, maple syrup urine disease; IP, intact protein; Phe, phenylalanine; PKU, phenylketonuria; RDI, recommended daily intake; TP, total protein; Tyr, tyrosine; TYR-1, tyrosinemia I; UCD, urea cycle defects.

## Data Availability

Not applicable.

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
