# Peer review of "Breastfeeding and Inborn Errors of Amino Acid and Protein Metabolism: A Spreadsheet to Calculate Optimal Intake of Human Milk and Disease-Specific Formulas"

_nutrients, 2023, doi:10.3390/nu15163566_

Round 1
Reviewer 1 Report
The submitted manuscript from Vitoria-Minana et al titled “Breastfeeding and inborn errors of amino acid and protein metabolism: a spreadsheet to calculate optimal intake of human milk and disease-specific formulas” presents recommendations and Excel spreadsheets for calculating human milk (HM) and medical formula (SF-AA-Prot) needs for nine specific inborn errors of metabolism for which use of human milk is possible in diet management.
Unfortunately, I am not able to recommend publication of this article in its current form. I have various reservations about how the recommended intake of HM is determined for some of these disorders. Examples include:
I do not believe that the amount of HM and SF-AA is initially determined based on the amount of fluid required by the infant. The first calculation in these disorders should be the amount of human milk required to meet an initial estimate of the mg/kg for the limiting amino acid (i.e. phenylalanine in PKU, leucine in MSUD, etc.) or the amount of protein (g/kg) for propionic/methylmalonic acidemia and urea cycle disorders. This is the limiting factor in all of these disorders and needs to be used as the basis for the remainder of the calculations. Yes, meeting fluid needs is important, but definitely not the place to start.
I would also argue that using the “mean” mg/kg or g/kg as the basis for calculating HM needs is a gross oversimplification for these disorders. Yes, it may be appropriate to use the mean in some cases for some disorders (PKU, TYR-1), but certainly not for all. For example, an infant with classical MSUD should not be started out at a leucine intake of 70 mg/kg. Selecting a mg/kg at the low end of the range (i.e. 40 mg/kg or even lower) would make much more sense given the fragile nature of those with classical forms of this disorder.
Using a prescribed volume of medical formula to limit the intake of HM (from the breast or pumped milk) works for some disorders, but not for all. It is again an oversimplification of the diet treatment for some of these disorders. An example here would be a UCD – even in a mild UCD, I doubt that you’d want to recommend a specific volume for an essential amino acid-based formula and then allow adlib intake of breast milk. My guess is that you don’t do it that way, but it is not clearly evident in the paper.
I also don’t think that you emphasize the need to use lab values as the primary determinant for adjusting HM and SF-AA volumes after the initial diet prescription is determined.
I also don’t think that prescribing a number of feeds is the best approach for treatment of some of these disorders, especially if the infant is healthy. As much as possible, adlib intake throughout a 24-hour period is preferred and calculating specific volumes for feedings is rarely needed.
Needs some work, but content is my larger concern rather than language.
Author Response
Thank you for your comments, please find attached the answers as a file.

Reviewer 2 Report
References: these are incorrectly cited in text as there are two references numbered as 1. References need to be re-numbered and corrected in text. Some references need to be replaced by more recent references such as the composition of human milk, propionate production in the intestine. The references in the excel tool do not match the main paper. HCU reference in tool is in Spanish, but ref 36 is cited in text.
The authors need to recognise that dietetic management practices vary worldwide for the 9 disorders discussed and intakes of natural protein and SF-AA could vary significantly. This is particularly important for MMA, PA and UCD as the authors cite the MMA/PA and UCD guidelines but the excel tool uses US guidelines which have different protein guidelines. The authors need to recognise the concerns about use of SF-AA in MMA and PA.
This paper provides an excel spreadsheet to calculate the amounts of SF-AA and HM for 9 IEM. The calculation is done manually rather than by an excel formula. The authors need to describe how to do the manual calculation or provide a formula in the excel programme
The authors need to provide evidence demonstrating the excel tool works in clinical practice by real case examples. This type of calculation is already used in clinical practice and benefit of using such a tool needs to be provided.
The authors need to recognise the role of biochemical monitoring in determining amount of human milk to give rather than theoretical requirements which can only be used as a guide. This emphasis the importance of including examples of clinical case the tool has been used for.
30 - The authors state this tool may help increase HM consumption but provide no evidence to support this.
45 - PKU European guidelines were cited in text but not included in the calculation tool.
51, 52 The authors should cite the first publications in breastfeeding in PKU and rationale for this.
53- intermediate is incorrect terminology - intermediary metabolism. The paper should use IEM. Line 83 - then refers to this as pathologies which is incorrect terminology.
Line 96 - mean values were used for the excel tool. This needs discussion as guidelines give a range suggesting requirements may differ across the 0-6months age range or severity of the disorder
Table 2 the legend needs to be written more clearly. TYR-1 error in abbreviation PT. some of the references are cited incorrectly eg HCU is 20 (which is MSUD) but this should probably be 36. The energy requirements range greatly for the different disorders and need explanation.
Table 3 - this needs further explanation as for some disorders 100% of intake could be HM and this does not match with guidance from Table 2. The authors comment on this but guidelines suggest this is not the case in clinical practice. In IVA the authors suggest the volume of HM tolerated is low but this is not the case in clinical practice. Later in the discussion this is recognised but the spreadsheet uses a restricted amount.
Some spelling errors in English in both paper and spread sheet (eg: spelling of protein).
Some repetition of text
Some sentences could be more concise
Author Response

(The authors gave the same response as above.)

Reviewer 3 Report
This study aimed to calculate the nutritional requirements and appropriate HM or special formula intake in infants with IEM-AA-P related diseases though designing a spreadsheet. This is an important topic, but further work still be needed. Please see my comments below.
1. Feeding is a key issue in the development of infants, which could be influenced by many factors, and it is often necessary to develop a personalized feeding program for infants with diseases. In view of the fact that the spreadsheet in the manuscript is only a preliminary calculation based on relevant guidelines and lacks confirmatory data, it is still a long way from becoming a tool for clinical use. At present, it is not certain that it has reference significance.
2. The background and discussion sections lack statement of the practice and use of existing feeding calculating tools.
3. Table 2 presents the recommended daily intake of limited essential amino acids or proteins, and energy for each IEM. But the theoretical total daily intake of HM (%) or special formula present in Table 3 and table 4 were calculated based on amino acid or proteins t only. How about the energy or other nutrients needed?
4. The current discussion part is more like the calculation basis of the data in the result section, which lacks the use of spreadsheet and discussion with other existing feeding related tools.
Author Response

(The authors gave the same response as above.)

Round 2
Reviewer 1 Report
The second version of the manuscript “Breastfeeding and inborn errors of amino acid……” is much improved compared to the first version. Your revisions to your spreadsheets to first calculate the amount of HM to meet the limited protein/amino acid goal really helps. However, I feel it needs more work before it is ready to publish.
Here are my observations.
Line 18, 60, 82 and likely other places in the manuscript: Change all “proteins” to protein.
Lines 28, 396, 432: extracted HM. Use the word “expressed” instead of “extracted”. Check your wording throughout to make sure you are using “expressed” milk.
Remove lines 36 and 37. I don’t think this statement is necessary. You are saying basically the same thing in the paragraph before this.
Line 81: What is “mixed lactation”. Need to reword to explain what you mean.
Line 118: Is FAA and SF-AAE spelled out in the manuscript before this point? (I’m not seeing them). Make sure that all your abbreviations are spelled out the first time they are used in the paper.
Table 2: For TYR-1, PT should be TP.
Line 135: Need to say where/how the reader can access the spreadsheets. Is there a website that can be accessed or are they in supplementary material? This needs to be stated so readers know where to look to find them.
Line 154: The g/kg values that you are giving are for intact protein, not total protein. Example: in UCD, 0.8 – 1.1 g/kg is the recommendation for intact protein. Total protein would be 1.2-2.2 g/kg. I noticed that these values were correctly addressed in the Discussion so perhaps they aren’t needed in this part of the paper. But change the wording to “intact protein” if you do leave it here.
Lines 175, 193: Your abbreviation is IEM-AA, not EIM-AA. Suggest checking all your abbreviations to make sure they are correct.
Lines 213 – 224: Is this paragraph necessary? It seems like what you are saying about PKU in other parts of the Discussion is enough.
Lines 248 – 256: I agree that we need to monitor various nutrition markers, but since your spreadsheets don’t include vitamins, minerals, etc. I don’t think the paragraph is necessary.
Line 297: Add the range for lysine (65 – 100 mg/kg). You only state 100 mg/kg in the sentence.
Line 378: Here you are saying “intact protein”. Other places in the manuscript, you use “natural protein”. Need to choose one and be consistent throughout.
Line 422: It is correct to say that the volume of HM is determined first. But, if you make up the remaining volume of the formula with SF-AAE, you may end up with an excessive Total Protein content. You need to meet both an Intact protein goal and a protein goal for essential amino acids. For UCD, it is very common to need to use energy modules (fat, CHO only) to make up needed calories because both HM and EAA formula volumes need to be restricted. Perhaps you can suggest calculating the total protein after you use your spreadsheet and if total protein is excessive, suggest limiting the amount of EAA formula you are adding to the formula mixture?
Line 450: What does “use with error” mean? This sentence needs to be reworded.
The Discussion is very long and needs to be condensed so that it is easier to read. I'm not sure all of the details for calculating specific diets for each disorder is really necessary.
This version is improved compared to first version. But, still could use some additional editing to make it read smoother. I did point out some places to change wording in my comments.
Author Response
Thank you for your review, which we appreciate as it allows us to improve our work.
In the attached document we include the modifications made to the article thanks to your comments.
Thank you for your review, which we appreciate as it allows us to improve our work.
In the following comments we detail the changes made to the text according to your suggestions.The current lines number are from version 3.
Line 18, 60, 82 and likely other places in the manuscript: Change all “proteins” to protein.
We have changed “proteins” to “protein” in the lines 19,63,77,107,108,126 (table 1), 129,164, 237,264 (deleted into the paragraph) and 414.
Lines 28, 396, 432: extracted HM. Use the word “expressed” instead of “extracted”. Check your wording throughout to make sure you are using “expressed” milk.
We have changed “extracted” to “expressed” in the lines 30 (expressed),407 (express), 431 (express),445 (expressed),474 (expressing)
Remove lines 36 and 37. I don’t think this statement is necessary. You are saying basically the same thing in the paragraph before this.
The text in the lines 38-39 are deleted…….
This tool allows for rapid calculation of nutritional requirements and appropriate HM intake in these IEM-AA-Ps, and may facilitate management of mixed lactation in affected patients.
Line 81: What is “mixed lactation”. Need to reword to explain what you mean.
The current text reads as follows in lines 83-84:
in order to facilitate management of mixed lactation and encourage breastfeeding in infants with IEMs.
The new text should read:
in order to facilitate management of mixed lactation and encourage breastfeeding in infants with IEMs.
Line 118: Is FAA and SF-AAE spelled out in the manuscript before this point? (I’m not seeing them). Make sure that all your abbreviations are spelled out the first time they are used in the paper.
Thank you for your comment, these two abbreviations appear for the first time in the text.
The current text reads as follows in the lines 121-122:
Next, the estimated total daily fluid intake is entered, and the amount of FAA mixture containing Met, Thr, Val, and Ile (for MMA/PA) or SF-AAE (for UCD),
The new text should read:
Next, the estimated total daily fluid intake is entered, and the amount of special formula without Met, Thr, Val, and Ile (SF-Met-Thr-Val-Ile) for MMA/PA or special formula with essential amino acids (SF-AAE) for UCD,
The current text reads as follows in lines 174 and 176:
FAA-Met-Thr-Val-Ile
The new text should read:
SF-Met-Thr-Val-Ile
We have revised the whole text and in this aspect, we need make these changes:
Lines 21,63,72,91,95,118,163,485……. (IEM-AA-Ps) is changed by (IEM-AA-P).
Line 151…. IEM-AA is changed by inborn errors of metabolism of amino acids (IEM-AAs)
Lines 215,221,223 and 458….. IEM-AA is changed by IEM-AAs
Table 2: For TYR-1, PT should be TP.
The text is changed
Line 135: Need to say where/how the reader can access the spreadsheets. Is there a website that can be accessed or are they in supplementary material? This needs to be stated so readers know where to look to find them.
Thank you for your suggestion.
In the current lines 145-146 we have added the text…
The spreadsheet is available online at www.mdpi.com/xxx/s1: Supplementary Data 1
Line 154: The g/kg values that you are giving are for intact protein, not total protein. Example: in UCD, 0.8 – 1.1 g/kg is the recommendation for intact protein. Total protein would be 1.2-2.2 g/kg. I noticed that these values were correctly addressed in the Discussion so perhaps they aren’t needed in this part of the paper. But change the wording to “intact protein” if you do leave it here.
In the line 165, “total protein” is changed by “intact protein”.
Lines 175, 193: Your abbreviation is IEM-AA, not EIM-AA. Suggest checking all your abbreviations to make sure they are correct.
In the lines 188 and 195, EIM-AA-Ps is changed by (IEM-AA-P)
In the line 207,we have changed EIM-AAs by IEM-AAs
In the lines 471 and 482,we have changed EIM-AA by IEM-AAs.
Lines 213 – 224: Is this paragraph necessary? It seems like what you are saying about PKU in other parts of the Discussion is enough.
The new lines 231-235 are deleted….
PKU is the IEM-AA for which there is the greatest abundance of data regarding the use of HM. Infants with blood Phe >360 μmol/L at diagnosis should be treated with a low-Phe diet. If the Phe concentration at diagnosis is >1000 μmol/L, only SF-Phe should be ad-ministered, and should result in a decrease of 400 μmol/L/day until reaching the desired values of 120–360 µmol/L. Therefore,
Lines 248 – 256: I agree that we need to monitor various nutrition markers, but since your spreadsheets don’t include vitamins, minerals, etc. I don’t think the paragraph is necessary.
The paragraph 262-270 is deleted
For PKU, as well as the other IEM-AAs included in the spreadsheet, nutritional control typically consists of measurement of multiple parameters, which commonly include hemogram, iron (Fe) levels, vitamin B12, folic acid, cholesterol, triglycerides, total proteins protein, albumin, prealbumin, calcium (Ca), phosphorus (P), vitamin D3, zinc, and selenium [36]. Assessment of these nutritional parameters provides important information about the patient's nutritional status and helps monitor the adequacy of various essential nutrients, minerals, and vitamins. By regularly monitoring these parameters, healthcare professionals can ensure appropriate nutritional support and make necessary adjustments to the dietary management of PKU and other IEM-AAs.
Line 297: Add the range for lysine (65 – 100 mg/kg). You only state 100 mg/kg in the sentence.
In the table 2 of the current guidelines cited in the article (19.- Boy, N.; Mühlhausen, C.; Maier, E.M.; Ballhausen, D.; Baumgartner, M.R.; Beblo, S.; Burgard P.; Chapman K.A.; Dobbelaere D.;Heringer-Seifert J.; et al. Recommendations for diagnosing and managing individuals with glutaric aciduria type 1: Third revision. J. Inherit. Metab. Dis. 2022. doi: 10.1002/jimd.12566. Epub ahead of print ) it is recommended an intake of 100 mg/kg/day of lysine for infants 0-6 months which is the age for which the spreadsheet is designed. Therefore, it seems to us that we should not change it in line 309.
Line 378: Here you are saying “intact protein”. Other places in the manuscript, you use “natural protein”. Need to choose one and be consistent throughout.
We have decided to replace “natural protein” by “intact protein” throughout the text……..
Lines 22,107,134 (and we change the abbreviature NP by IP in table 2), 197,211, 237,280,320, 360,406,413, 416,461 and 472.
In the spreadsheed we have changed “whole protein requirements” by “intact protein requirements” in MMA/PA and UCD sheets.
Line 422: It is correct to say that the volume of HM is determined first. But, if you make up the remaining volume of the formula with SF-AAE, you may end up with an excessive Total Protein content. You need to meet both an Intact protein goal and a protein goal for essential amino acids. For UCD, it is very common to need to use energy modules (fat, CHO only) to make up needed calories because both HM and EAA formula volumes need to be restricted. Perhaps you can suggest calculating the total protein after you use your spreadsheet and if total protein is excessive, suggest limiting the amount of EAA formula you are adding to the formula mixture?
We think your comment is very appropiate as we also had doubts about allowing them to take too much SF-AAEs. We therefore thank you for your idea and add the following text to line 432.
We suggest calculating the total protein with the spreadsheet and if it is excessive, supplementing the energy needs with lipid and/or carbohydrate modules, thus avoiding excessive SF-AAE intake.
Line 450: What does “use with error” mean? This sentence needs to be reworded.
In line 464 “with error” is deleted
The Discussion is very long and needs to be condensed so that it is easier to read. I'm not sure all of the details for calculating specific diets for each disorder is really necessary.
We have deleted these paragraphs
Lines 283-285 (tyrosinemia I)
The introduction of regular feeding with HM or IF should be adjusted to achieve a Tyr intake of 95–275 mg/day in the case of TYR-1.
Lines 310-311 (GA I)
A fluid intake value between 120 and 200 ml/kg/d can be inputted.
Lines 343-345 (HCU)
A fluid intake value between 120 and 200 ml/kg/d can be inputted.
Lines 351-355 (MSUD)
Based on these dietary recommendations, the use of HM as a source of intact protein (including Val, Leu, and Ile) should be considered in the dietary management of infants with MSUD, provided that frequent anthropometric, clinical, and laboratory follow-up is performed. This includes nutritional analysis and measurement of plasma levels of Val, Leu, and Ile
Lines 369-372 (MSUD)
A survey by McDonald et al. [46] documented 17 infants with MSUD who were breastfed on demand in combination with SF-AA, but did not include detailed information on patient health status and the occurrence of metabolic decompensations.
Line 387 (IVA)
In the survey by McDonald et al. [46], 13 infants with IVA were provided with HM.
Lines 420-421 (UCD)
A review reported that 58% of UCD patients continued with HM at 16 weeks of age, but follow-up information was not provided [46].

Reviewer 3 Report
Thank you for addressing all the comments.
Author Response
Thank you for your review as it has helped us a lot to improve the article.